# Seasonal $M_2$ Internal Tides in the Arabian Sea

**Jingyi Ma** , **Daquan Guo \***, **Peng Zhan** and **Ibrahim Hoteit**

Division of Physical Sciences and Engineering, King Abdullah University of Science and Technology, Thuwal 23955, Saudi Arabia; jingyi.ma@kaust.edu.sa (J.M.); peng.zhan@kaust.edu.sa (P.Z.); ibrahim.hoteit@kaust.edu.sa (I.H.)
\* Correspondence: daquan.guo@kaust.edu.sa

**Abstract:** Internal tides play a crucial role in ocean mixing. To explore the seasonal features of mode-1 $M_2$ internal tides in the Arabian Sea, we analyzed their propagation and energy distribution using along-track sea-level anomaly data collected by satellite altimeters. We identified four primary source regions of internal tides: Abd al Kuri Island, the Carlsberg Ridge, the northeastern Arabian Sea, and the Maldive Islands. The baroclinic signals that originate from Abd al Kuri Island propagate meridionally, whereas those originating from the west coast of India propagate southwestward. The strength and energy flux of the internal tides in the Arabian Sea exhibit significant seasonal and spatial variability. The internal tides generated during winter are more energetic and can propagate further than those generated in summer. Doppler shifting and horizontal variations in stratification can explain the differences in the internal tides' seasonal distributions.

**Keywords:** internal tides; energy flux; satellite; Arabian Sea

## 1. Introduction

Internal tides, also known as baroclinic tides with tidal frequencies, may lead to prominent preconditioning of ocean mixing [1]. Barotropic tidal energy is converted to baroclinic tidal energy, which mainly occurs in regions exhibiting rough topography where disturbances are generated, such as sea mountains, ridges, and trenches [2–4]. Baroclinic tidal energy can propagate over distances of several thousand kilometers from the source region [5,6]. Internal tides play an important role in both physical and biochemical processes in the ocean. For instance, internal tides could be one of the driving forces of the Meridional Overturning Circulation (MOC), which is a major component of the climate system [7,8]. Further, internal tides can promote ocean mixing, thus connecting the deep ocean with the entire Earth system [3]. They can also transport the frequent pulses of nutrients, which cause algal blooms in shallow depths, the areas that are generally more nutrient-limited than the deep ocean [9].

As internal tides are generated in a stratified ocean, their properties are influenced by certain parameters related to the background ocean environment, such as temperature, salinity, and background currents. Variations in these water parameters can affect the generation and propagation of internal tides [10,11]. Through numerical simulations, Zaron and Egbert [12] revealed that the variability of internal tides generated at the Hawaiian Ridge is associated with the variation of an ensemble of background fields along the propagation pathways of these waves. Using a three-dimensional (3D) ray-tracing method, Chavanne et al. [13] studied the properties of internal tides in the Kauai Channel, Hawaii. They showed that the inhomogeneous buoyancy frequency field and the Doppler-shifting effect are the most important and second-most important factors influencing the energy of internal tides. Müller et al. [14] examined the seasonality of the global $M_2$ internal tides via a numerical simulation and revealed the effect of varying seasonal stratification on internal tides. Similarly, Jeon et al. [15] implemented a numerical model to analyze semidiurnal internal tides in the East/Japan Sea and suggested that variations in the stratification can

influence the wavelength of internal tides and eventually control their propagation distance and energy dissipation. These aforementioned studies suggest that seasonally varying background stratification and currents significantly impact the properties of internal tides.

The properties of internal tides have also been studied through in situ observations and satellite datasets. In situ observation provides information on the vertical characteristics of the internal waves (e.g., the energy flux [16]); however, these observations are generally too scarce in the space and time domains to obtain a systematic understanding of the baroclinic tides. In contrast, satellite altimeter measurements have the advantages of a wider orbit and longer temporal coverage, enabling the detection of internal tides across a much broader domain than the few locations covered by in situ observations [17]. Hence, satellite altimeter measurements are now widely used to monitor internal tides across the global ocean (e.g., the Hawaiian Ridge [18], the South China Sea [19], and the Tasman Sea [20]). Inference methods of the internal tides and the quality of the generated data from satellites have considerably improved in recent years, which has facilitated the use of satellite remote sensing methods to study internal tides [6,21–23]. Ray and Zaron [19] demonstrated the feasibility of extracting seasonal baroclinic signals from altimeter data in the Pacific, thus providing a basis for studying seasonal internal tides using altimeter data. By combining data from multiple satellites, Zhao et al. [24] achieved higher spatial and temporal resolution than that achieved using data obtained from a single satellite; thus, using datasets obtained from multiple satellites is a more effective approach than using datasets obtained from a single satellite for investigating a specific basin.

The Arabian Sea is located in the north of the Indian Ocean; it is bounded on the north by the Gulf of Oman and on the west by the Gulf of Aden. The general circulation in this basin is clockwise in summer and anticlockwise in winter [25], and it exhibits remarkable seasonal variability driven by the Indian monsoon system [26,27]. The eddy-induced swirl transport is larger in the western Arabian Sea and tends to compensate for heat transport by the mean flow [28]. The two gulfs in the north connect and exchange water properties with the Arabian Sea through two straits [29]. The seasonal variations in stratification possibly influence the properties of internal tides in this basin. Previous studies on internal tides in the Arabian Sea were markedly limited and mostly based on in situ observations [30,31]. However, due to the development of global satellite observation networks, several studies recently indicated that the Arabian Sea is a hotspot of internal tides [24,32].

In situ data revealed that temperature fluctuations caused by internal tides are related to a thermocline gradient in the northeast Arabian Sea [33]. Data from acoustic Doppler current profilers (ADCPs) were analyzed to identify strong internal tides generated on the continental shelf of the eastern Arabian Sea; the strong seasonal stratification on the shelf is believed to generate large semidiurnal internal tides during the southwest monsoon [34]. Subeesh et al. [35] performed numerical simulations to investigate the properties of internal tides on the shelf-slope off the west coast of India. They found that about 2.4 GW of barotropic tides are converted to baroclinic tides in this region. However, the above-mentioned studies exclusively focused on the internal tides in specific areas of the Arabian Sea. The general characteristics of the internal tides in the Arabian Sea, such as their spatial distribution, generation sources, propagation paths, and dissipation areas, as well as their seasonal variations and mechanisms have not been examined thus far and are poorly understood.

In this study, data from all available satellite altimeters were used to analyze the internal tides and describe their properties in the Arabian Sea. Specifically, we detected and reproduced surface mode-1 $M_2$ internal tides. Using the obtained the data, we could provide a general description of the internal tides' characteristics, including their sources, propagation directions, and integrated energy flux and seasonal variations. We further analyzed the mechanisms that drive the seasonal variation in the internal tides with respect to the superposition of waves and horizontal changes in stratification. The remainder of this paper is organized as follows. Section 2 describes the data and methods. Section 3 presents the main results of the characteristics of the internal tides and their

seasonal variability. A general discussion is provided in Section 4. Finally, Section 5 summarizes the main conclusions.

## 2. Data and Methods

### 2.1. Data

Sea-level anomaly (SLA) along-track measurements obtained from eight satellites—European Remote Sensing 1 (ERS-1), European Remote Sensing 2 (ERS-2), Environmental Satellite (Envisat), Geosat Follow-On (GFO), Topex/Poseidon, Jason-1, Jason-2, and Cryosat-2—and their tandem missions were used to extract the features of mode-1 $M_2$ internal tides in the Arabian Sea. All datasets were obtained from the Online Data Extraction Service (ODES) of the Archiving, Validation, and Interpretation of Satellite Oceanographic Data (AVISO) project and are available online. The aforementioned satellites have different data coverage. For instance, Jason-1 has a repeat cycle of 9.9156 days with 254 passes per cycle. The repeat cycle of Cryosat-2 is 369 days with 30-day pseudo-subcycles and ground-track drift within a 5 km band. Cryosat-2 provides the densest track coverage.

The merged satellite data thus obtained provide extensive spatial coverage, as shown in Figure 1a. We analyzed the satellite data for the area within [0°, 30°N] and [30°E, 80°E] from January 1993 to January 2016. Figure 1b shows the topography of the study domain, including the northern Indian Ocean, Arabian (or Persian) Gulf, and the Red Sea.

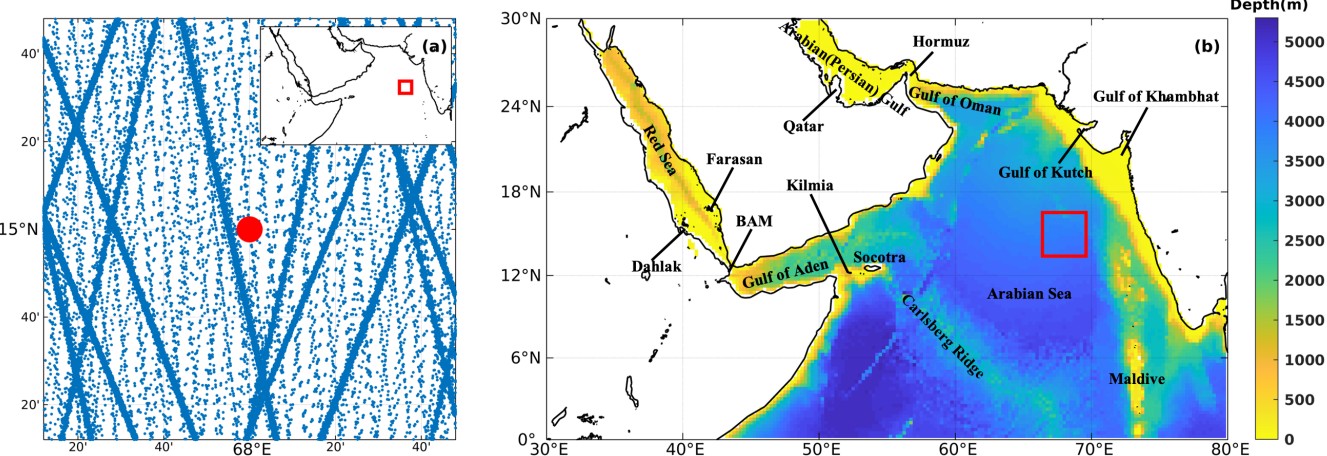

**Figure 1.** (**a**) Spatial distribution of satellites' sampling. The position of the selected window is shown on the top right corner, indicated by the red box. Blue dots show the satellite observation points, and the "line segments" actually are built up by dense dots. The red point is the central point of this window. (**b**) Topography of the Arabian Sea. The main locations involved in this study are indicated.

### 2.2. Methods

The process of extracting internal tides from satellite data involves two main steps: data preprocessing and plane-wave fitting [18]. The purpose of the preprocessing step is to remove potential outliers and barotropic signals from the original data and calculate the parameters that will be used in the next step. Initially, all values along one track greater than 3.5-times the standard deviation and those measured at water depths less than 50 m were tagged as outliers and thus removed from the dataset. To extract the baroclinic signals from the original satellite data, the barotropic component predicted by the TOPEX/POSEIDON global tidal model (TPXO9) was subtracted. The wavelengths of the $M_2$ barotropic and baroclinic tides are about 1000 and 100 km, respectively. Therefore, they can be distinguished using a spatial bandpass filter designed as a fourth-order Butterworth filter, considering their wavelengths. The thresholds were selected based on the wavelengths of the mode-1 $M_2$ internal tides in the Arabian Sea as 50 km and 200 km. The wavelengths were computed by Equation (1) [23]:

$$\omega^2 = k^2 c^2 + f^2. \tag{1}$$

This equation describes the dispersion relation of waves under rotation, where $\omega$ represents the tidal frequency, $k$ represents the wavelength, $c$ represents the phase speed, and $f$ represents the Coriolis frequency ($f \equiv 2\Omega\sin(\varphi)$, with $\varphi$ denoting the latitude).

Phase speed $c$ was calculated using the World Ocean Atlas (WOA) 2018 data and the following relation [23,36]:

$$\frac{d^2\Phi(z)}{dz^2} + \frac{N^2(z)}{c^2}\Phi(z) = 0, \tag{2}$$

where the vertical structures of displacement and vertical velocity are described by $\Phi(z)$, which can be calculated by the eigenvalue of Equation (2), subject to the boundary conditions $\Phi(0) = \Phi(-H) = 0$ (H expresses the water depth), and $N(z)$ expresses the buoyancy frequency profile. The Brunt–Vaisala frequency is defined as $N^2 = -(g/\rho_0)d\rho/dz$, where $\rho$ represents density and is calculated from the WOA dataset. Following these preprocessing steps, the data were used in the below plane-wave-fitting analysis.

This research used the plane-wave-fit method proposed by Zhao and Alford [23] to analyze internal tides. The main idea of this method is to separate the waves propagating to different directions at a given location (e.g., the red point in Figure 1a). After estimating $k$ using Equation (1), the plane-wave-fit method [18] was applied to extract the features of the internal tides at the central point of one fitting window (e.g., Figure 1a), including their amplitudes, phases, and propagation directions. This method is similar to harmonic analysis, and both are based on the least-squares approach. Compared with harmonic analysis, which only uses a series of data at a given location, all data points within a $1.6° \times 1.6°$ window were included in the plane-wave-fit calculation:

$$SSH(x, y, t) = \sum_{i=1}^{3} a_i \cos(\omega t + \varphi_i - kx\cos\theta_i - ky\sin\theta_i), \tag{3}$$

where $SSH(x, y, t)$ represents the contribution of internal tides to the sea surface height (SSH). Here, $\omega$ and $k$ represent the frequency and wavenumber of $M_2$ internal tides, respectively, and $k$ can be obtained from Equation (1). Further, $a_i$ and $\varphi_i$ represent the amplitude and phase, respectively, which are the unknown variables in Equation (3), and $x$ and $y$ represent the east and north coordinates, respectively. Time is represented by $t$. Additionally, $\theta_i$ refers to the direction in one fitting window and ranges from $1°$ to $360°$ with an interval of $1°$. The letter $i$ represents the number of internal tides considered within the window. The first three largest internal tides ($i = 3$) were extracted, considering the fourth $M_2$ internal tides in the region to be insignificant. If readers are interested in this method, more details can be found in Zhao and Alford [23], Zhao et al. [24]

An example is shown in Figure 2, describing the relationship between the amplitudes of $M_2$ internal tides and their propagation directions for the fitting window ranging from $14.2°$N to $15.8°$N and $67.2°$E to $68.8°$E. The first step was to divide the window into $360°$ degrees and apply Equation (3) over the whole window data for each degree. Thus, each angle corresponds to one fitting with an amplitude $a_i$ and a phase $\varphi_i$. Figure 2a outlines the relationship between angle and amplitude, which represents the three most significant internal tides. The most prominent signal in this window appears along $225°$ (Wave 1), which is regarded as the largest internal tide at ($15°$N, $68°$E). The second step is to extract the second largest internal tides (Wave 2) from Figure 2a. Note that the influence of Wave 1 is observed not only along $225°$, but also along other directions, so the contribution of Wave 1 must be removed. Equation (3) was used again to predict the SSH generated by Wave 1, considering the amplitude and phase of Wave 1 were known from Figure 2a. Then this prediction was subtracted from the original data, as shown in Figure 2b. The third step is to repeat the above process to extract the third significant wave, as shown in Figure 2c. After completing the calculation for one fitting window, these operations are extended to the entire domain with a step of $0.25°$.

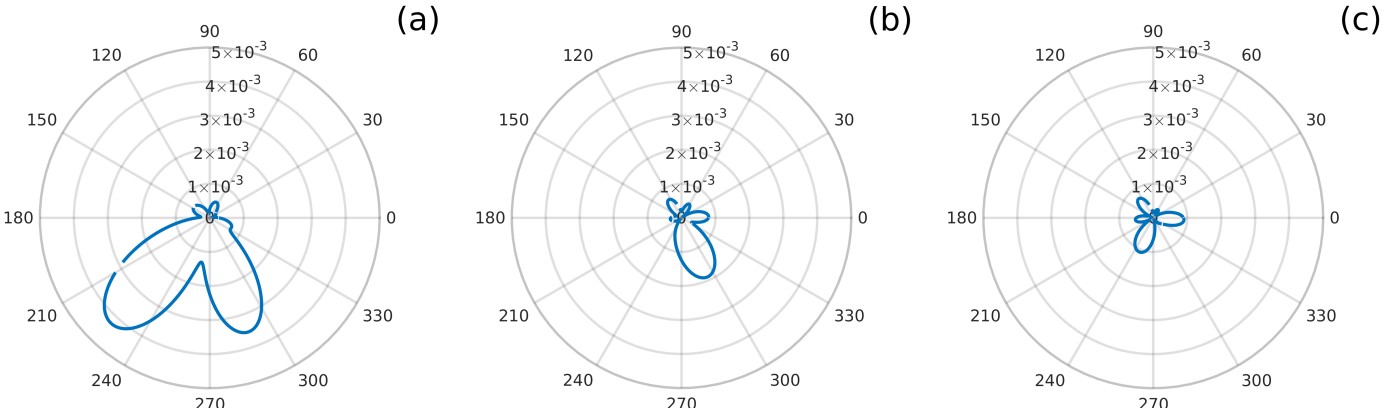

**Figure 2.** Amplitude (m) versus direction at (15°N, 68°E) as obtained from the plane-wave-fitting method within in a 1.6° × 1.6° window. Each lobe represents an internal wave. (**a**) All internal waves at (15°N, 68°E). (**b**) The residual waves after removing the largest lobe. (**c**) The residual waves after removing the first- and second-largest lobes.

Finally, the energy flux was calculated following the works of Pedlosky [37], Chiswell [38], and Zhao and Alford [23]. The vertical structures of baroclinic pressure and horizontal velocity, $\Pi(z)$, were first calculated using:

$$\Pi(z) = \rho_0 c^2 \frac{d\Phi(z)}{dz}, \tag{4}$$

where $\rho_0$ represents the density of seawater. The vertically integrated energy flux was then obtained as [24]:

$$
\begin{aligned}
F &= \frac{1}{2} \int_{-H}^{0} u(z) p(z) dz, \\
&= \frac{1}{2} a^2 \left[ \frac{\rho_0 g^2 \omega k_1}{\omega^2 - f^2} \int_{-H}^{0} \Pi^2(z) dz \right], \\
&= \frac{1}{2} a^2 F_n,
\end{aligned}
\tag{5}
$$

where $u(z)$ denotes the horizontal velocity; $p(z)$ denotes the pressure; $F$ denotes the vertically integrated energy flux; and $F_n$ denotes the integrated energy flux per unit amplitude.

## 3. Results

### 3.1. General Characteristics and Energy Flux

#### 3.1.1. Surface Displacement Due to Internal Tides

In this section, we first describe the general characteristics of the internal tides in the Arabian Sea and then examine their seasonal variabilities. A series of hourly snapshots of SSH revealed the generation and propagation features of mode-1 $M_2$ internal tides in this region. One such snapshot is shown in Figure 3, which presents the distribution of the SSH caused by internal tides at 00:53:20 on 5 November 2015. The hourly snapshots show the existence of several internal tides' sources scattered across the study domain, including the eastern Arabian Sea, Abd al Kuri Island, the Arabian (or Persian) Gulf, and the Red Sea. The amplitude of internal tides is larger in the open Arabian Sea, particularly in the northeastern part, compared with those in the Red Sea and the Arabian (or Persian) Gulf.



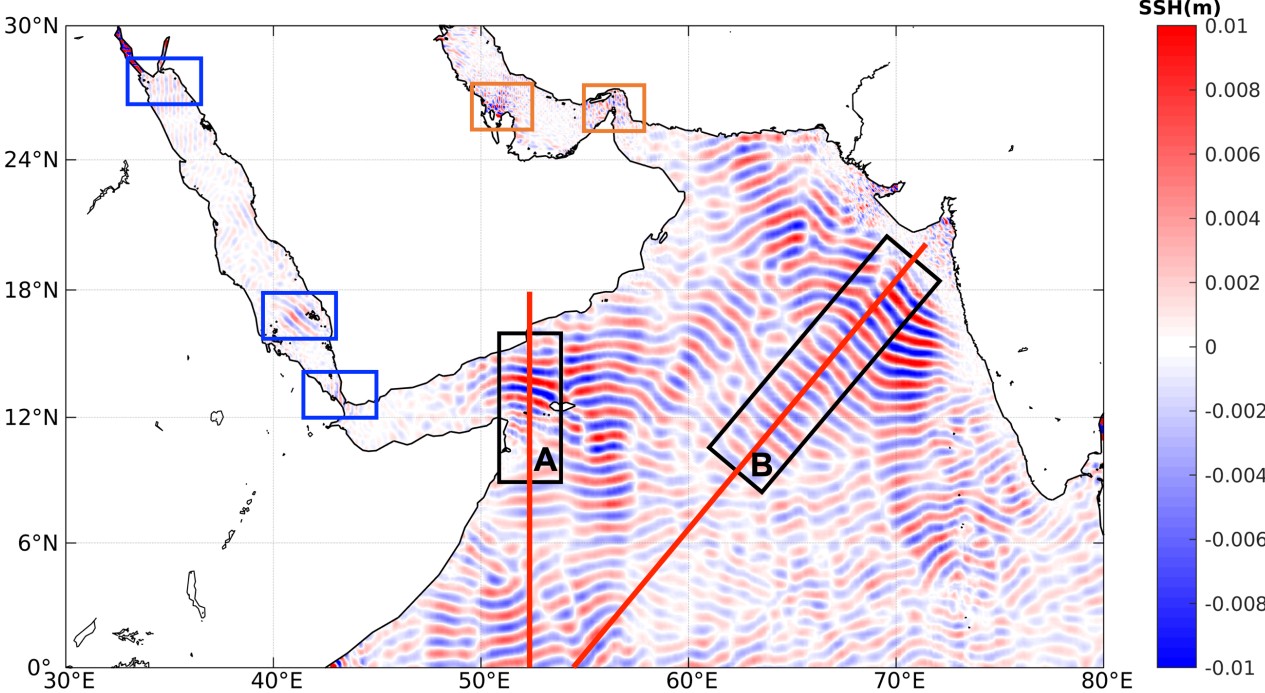

**Figure 3.** Snapshot of mode-1 $M_2$ internal tides' SSH at 00:53:20, 5 November 2015. Two primary generation sources of internal tides are suggested: Kilmia Island and the western India Peninsula, indicated by Rectangles A and B. The color represents the positive and negative values of the SSH. The three blue boxes indicate the generation locations of internal tides in the Red Sea. The red lines represent the locations of the profiles shown in Figure 4 along the internal tides' propagation direction.

In the Arabian Sea, the internal tides are mainly generated at Abd al Kuri Island, the Carlsberg Ridge, the northeastern Arabian Sea, and the Maldive Islands. The waves generated at Abd al Kuri Island (Box A in Figure 3) propagate both northward and southward. The northward waves, with an SSH ranging from −1.28 to 1.32 cm, are significantly more noticeable than the southward waves and exhibit more extensive coverage and a longer lifetime, reflected by a 200 km-wide crest line. The southward internal tides are more difficult to observe in the images, and they seem to rapidly dissipate after generation. More details regarding the southward internal tides are provided in Section 3.2. The Carlsberg Ridge, which is close to Abd al Kuri Island, also presents strong signals. As can be observed in the snapshot, the source location is around 10°N and features the sea ridge of the underwater topography. Moreover, the southward waves are weaker than the northward waves, with the former propagating over shorter distances and exhibiting a smaller SSH amplitude. Northward signals generated in the southern region beyond the study area may interact and suppress the southward internal tides generated from the Carlsberg Ridge.

The northeastern region of the Arabian Sea, which has an extensive continental shelf, is another source of internal tides. As shown in Figure 3, the internal tides generated in this region have two main branches: one that propagates southwesterly to the center of the Arabian Sea (Branch 1) and the other that propagates nearly southward (Branch 2). When the northeastward barotropic tides pass over the shelf break, the perturbation triggered by the topography develops into baroclinic tides that propagate offshore in the southwesterly direction forming Branch 1. Kumar et al. [33] identified similar characteristics of internal tides from in situ observations. Internal tides generated in this region can propagate over more than 1000 km (Box B in Figure 3). Moreover, their SSH ranges between −1.18 and 1.18 cm, suggesting that they are weaker than those generated at Abd al Kuri Island. When the southward waves (Branch 2) generated in the northeastern region of the Arabian Sea meet the northward waves generated at the Maldive Islands, strong superposition

occurs. Note that the east–west internal tides are suppressed as our analysis was based on filtered satellite along-track data [24], considering that the tracks are generally along the south–north direction.

In the Red Sea, we identified three main sources of internal tides as indicated by the blue rectangles in Figure 3: the northern Red Sea, southern Red Sea, and Strait of Bab-el-Mandeb (BAM); these locations are in agreement with those identified in previous studies [39–41]. A series of hourly snapshots reveals more details about the generation sites and propagation directions (not shown). Specifically, in the northern Red Sea, internal tides propagate into the Gulf of Suez and the Gulf of Aqaba. In the southern Red Sea, they are generated at Dahlak Island and propagate toward the northeast until they reach the coastline. In BAM, internal tides are generated when barotropic tides pass through the strait; the generated internal tides have an amplitude about five-times that of the tides in the southern Red Sea. Although internal tides in the southern Red Sea are weaker in terms of the amplitude, they can propagate over more than 300 km, which is a much wider range than those of the internal tides generated in the northern Red Sea and BAM. In the Arabian (or Persian) Gulf, we identified two sources of internal tides near the Strait of Hormuz and northern Qatar (as indicated by the two orange boxes in Figure 3). However, because of the shallow water depth and complex coastline, internal tides derived from satellite data may be contaminated by large uncertainties in the case of this basin.

After providing a general description of the internal tides in the study domain, we focus now on studying the propagation of the internal tides in specific directions. Figure 4a–f depicts the amplitudes and phases of the internal tides with the bottom topography along the two red Lines A and B highlighted in Figure 3. Instead of a superposition of the first three waves, we considered only the maximum wave because the superposition of phases is insignificant, and the maximum wave describes the major features of the internal tides.

The phase distribution along Line A is shown in Figure 4a. Here, two black arrows indicate the northward direction of propagation, in which case, the phase gradually increases from 0° to 360° and then sharply drops to 0. However, not all the phase falls rapidly. Although only the maximum wave was considered, the waves along Line A may have originated from different sources. Figure 4b,c shows the amplitude and bathymetry along the same transect. Note that the gray shading represents the area with significant internal tides, located between 11°N and 15°N and peaking around 13°N. Similarly, Figure 4d–f illustrates the same properties along the red Line B. The amplitude along this line reaches 0.77 cm at 17°N, and the consistency between the steepness of the topography and the amplitude suggests that the internal tides are generated from the area with steep topography.

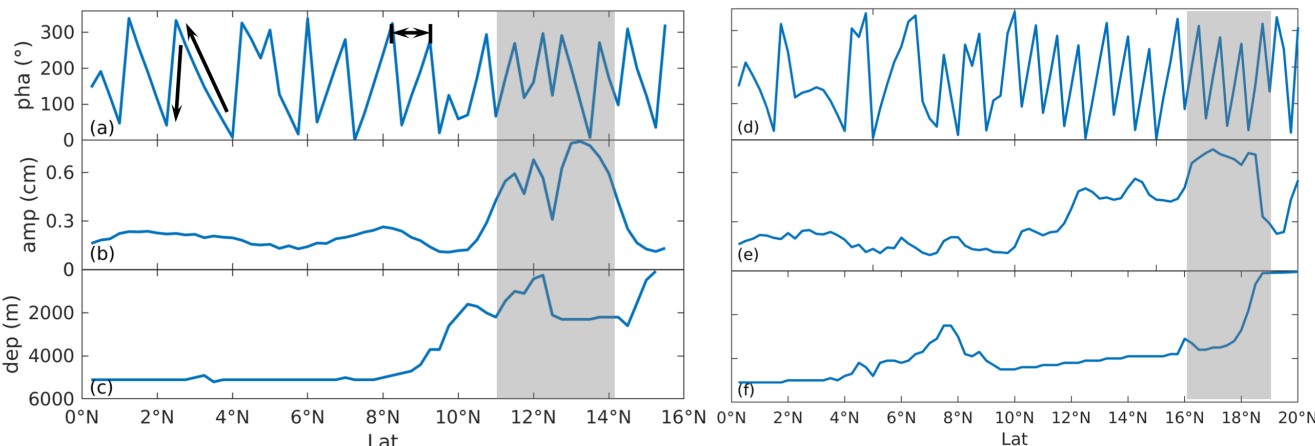

**Figure 4.** Changes of mode-1 $M_2$ internal tides along the red Lines A and B in Figure 3. (**a**,**d**) Internal tides' phase versus latitude. The propagation direction could be identified from the distribution of the phase, as indicated by the arrow. (**b**,**e**) Internal tides' amplitude versus latitude. Amplitude here is the displacement in sea surface caused by internal tides. (**c**,**f**) Water depth versus latitude. The gray shading area outlines the area with strong internal tides.

### 3.1.2. Energy Density and Flux

The distribution of the surface amplitude induced by the largest $M_2$ internal tides in three waves is shown in Figure 5a. The locations with a significant amplitude are almost consistent with the hotspots revealed in the SSH data (Section 3.1), with the maximum amplitude reaching up to 1.13 cm in the northeastern Arabian Sea. In addition, amplitudes near the Maldive Islands and the Carlsberg Ridge are around 0.5 cm, which is relatively higher than their surroundings. As the energy is proportional to the square of the amplitude, the vertically integrated energy flux is illustrated in Figure 5b. This value was calculated via the plane-wave-fitting method using the first three largest waves. Those three waves can propagate along different, even opposite, directions, hence the unorganized pattern in Figure 5b. The locations of the sources of the internal tides are distinct, and the magnitude and directions of the energy fluxes generally agree with the findings from the hourly snapshots (Section 3.1), revealing a reasonable consistency between the SSH and the energy flux. Moreover, the maximum energy flux reaches up to 10 kW/m in the northeastern Arabian Sea, and this value is comparable with that recorded around the islands of Hawaii [18], where internal tides are significant on a global scale.

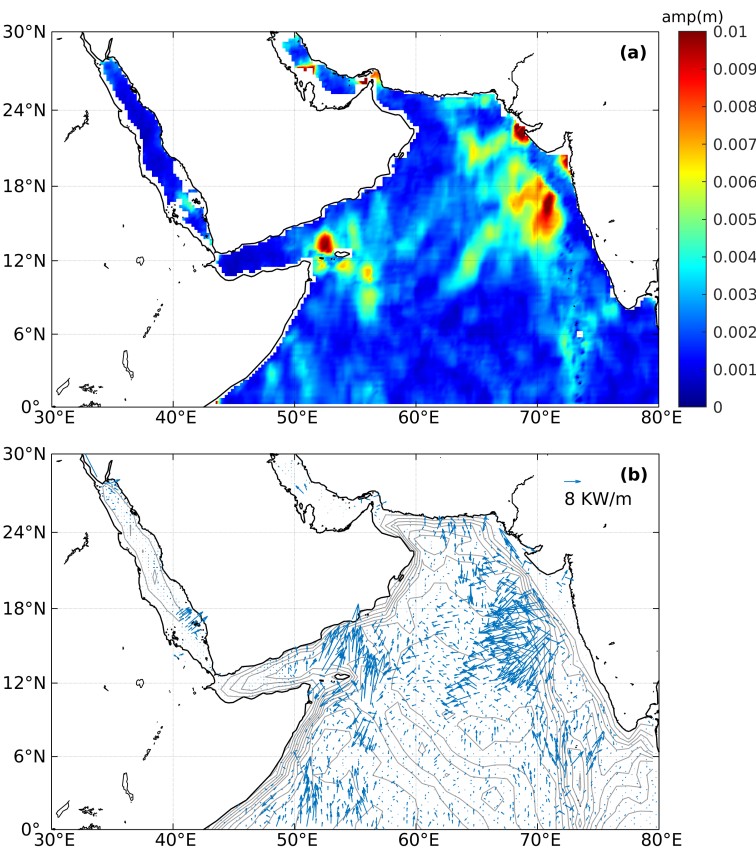

**Figure 5.** (**a**) Spatial distribution of $M_2$ internal tides' sea surface amplitude. (**b**) The vertically integrated energy flux of the mode-1 $M_2$ internal tides. Gray contouring in the background illustrates the bathymetry.

In the northern Red Sea, the energy flux generally propagates in an easterly direction at around 2 kW/m, which is much weaker than the energy flux in the hotspot locations of the Arabian Sea. In the southern Red Sea, the baroclinic waves are generated between the Farasan Islands and the Dahlak Islands, which then propagate northeast until reaching the coastline. Further, southeastward signals are observed to the south of the Farasan Islands, probably associated with the topography: when internal tides reach the Farasan Islands, the signals are split in two by the topography, with one moving toward the northeast

and the other moving toward the southeast. Similar phenomena were reported in the southern Red Sea using numerical simulations [41,42] and satellite synthetic aperture radar (SAR) imagery [39]. Some common features of internal tides can be extracted from both Figures 3 and 5b. The source locations of baroclinic tides have a high-gradient topography, which favors the generation of internal tides [1].

### 3.2. Seasonality of Internal Tides

To explore the seasonal variability of internal tides, the altimeter dataset was divided into summer (from July to September) and winter (from January to March) subsets, following the work of Ray and Zaron [19]. The corresponding seasonal stratification was extracted from the WOA dataset.

#### 3.2.1. Surface Displacement

Figure 6 shows two snapshots of the SSH of mode-1 $M_2$ internal tides in the Arabian Sea in (a) summer and (b) winter. In contrast with the general snapshot, larger values are observed in both summer and winter (Figure 3). The locations of the four main sources of internal tides are recognizable: Abd al Kuri Island, the Carlsberg Ridge, the northeastern Arabian Sea, and the Maldive Islands, which exhibit striking seasonal differences. During summer, there are almost no internal tides at the south of Abd al Kuri Island and the Carlsberg Ridge. Moreover, the northward internal tides generated at Abd al Kuri Island and the Carlsberg Ridge are still visible, but the signals seem more random than those observed in Figure 3. The situation in this region is completely opposite in winter: all the waves are strengthened, regardless of their direction. This is attributed to the background ocean environment and will be discussed further in Section 4.

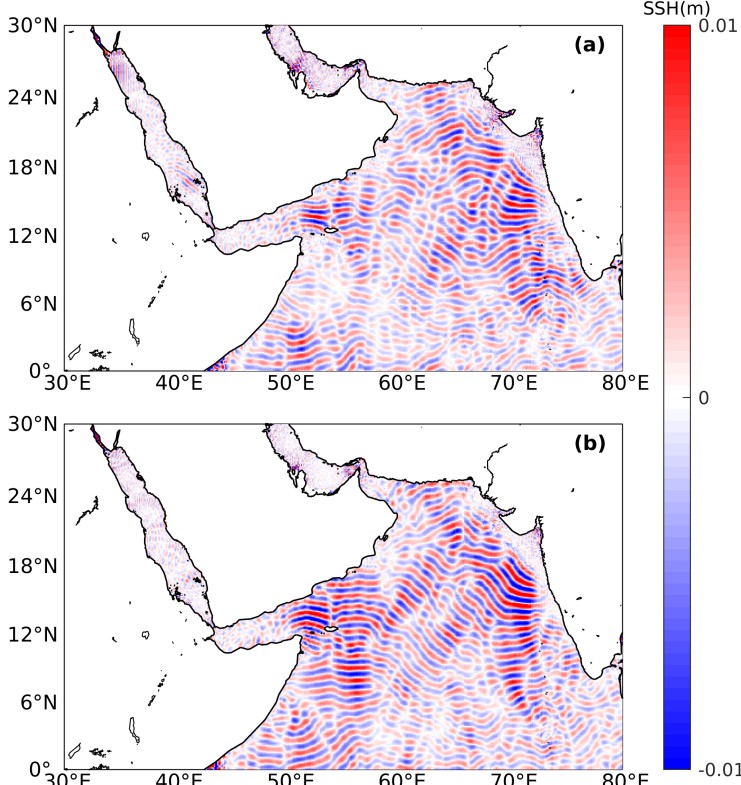

**Figure 6.** Seasonal snapshots of mode-1 $M_2$ internal tides' SSH at 00:53:20, 5 November 2015 in the Arabian Seas in (**a**) summer and (**b**) winter.

In the northeast Arabian Sea, the internal tides in winter are stronger. Branch 1 has a clear propagation routine in winter, but several waves following other directions mix with this branch in summer. The signals in winter have larger SSH and can propagate up to 6°N,

whereas they exhibit obvious incoherence between 12°N and 16°N in summer. In addition, Branch 2, which originates from the Northeast Arabian Sea, becomes significant in winter. Besides, the internal tides near the Maldive Islands travel over a longer propagation distance in summer.

The marginal seas also demonstrate pronounced seasonal variations. Internal tides in the northern Red Sea are strong and become significant at BAM during summer. The internal tides in the southern Red Sea are more organized in summer than in winter, probably owing to the superposition of internal tides from different directions. In the Arabian (or Persian) Gulf, the internal tides in summer are stronger than those in winter: even the source, which is close to Qatar, cannot be identified during winter.

### 3.2.2. Energy Flux

We further analyzed the seasonal variability of the energy flux intensity of internal tides in the Arabian Sea. The resulting vertically integrated energy flux is shown in Figure 7. Generally, the energy flux in winter tends to be larger than that in summer (Figure 3), with larger areas of a high-energy flux, including the northeastern Arabian Sea, the Carlsberg Ridge, and Abd al Kuri Island. One of the most significant difference is observed in the case of the northeastern Arabian Sea. During winter, the area with a high-energy flux is significantly larger than that in summer, and the stick-like pattern extends southwestward into the central basin; this pattern is not observed in summer. This winter pattern is also noticeable in the energy flux calculated using the annual data (Figure 5b), suggesting that winter internal tides significantly contribute to the annual energy flux.

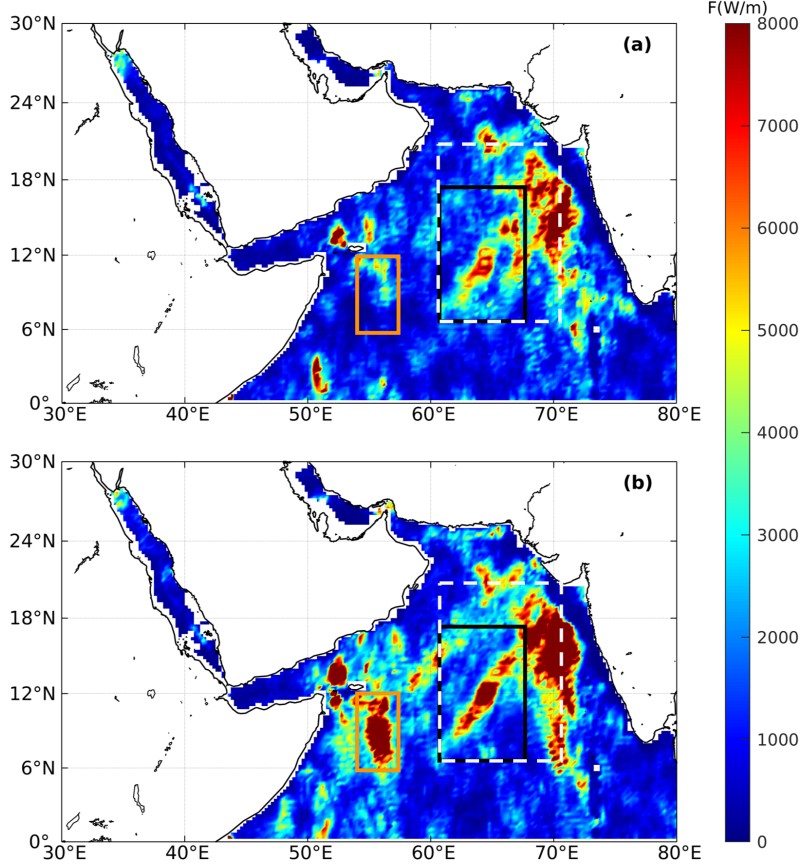

**Figure 7.** Long-term average vertically integrated energy flux in (**a**) summer and (**b**) winter, respectively. There are more hotspots during winter: the northeastern Arabian Sea, the Maldive Islands, the Gulf of Oman, and the southern Red Sea. The black box indicates the area of Figure 8a–c; the orange box indicates the area of Figure 8d–f; and the white box indicates the area of Figure 9.

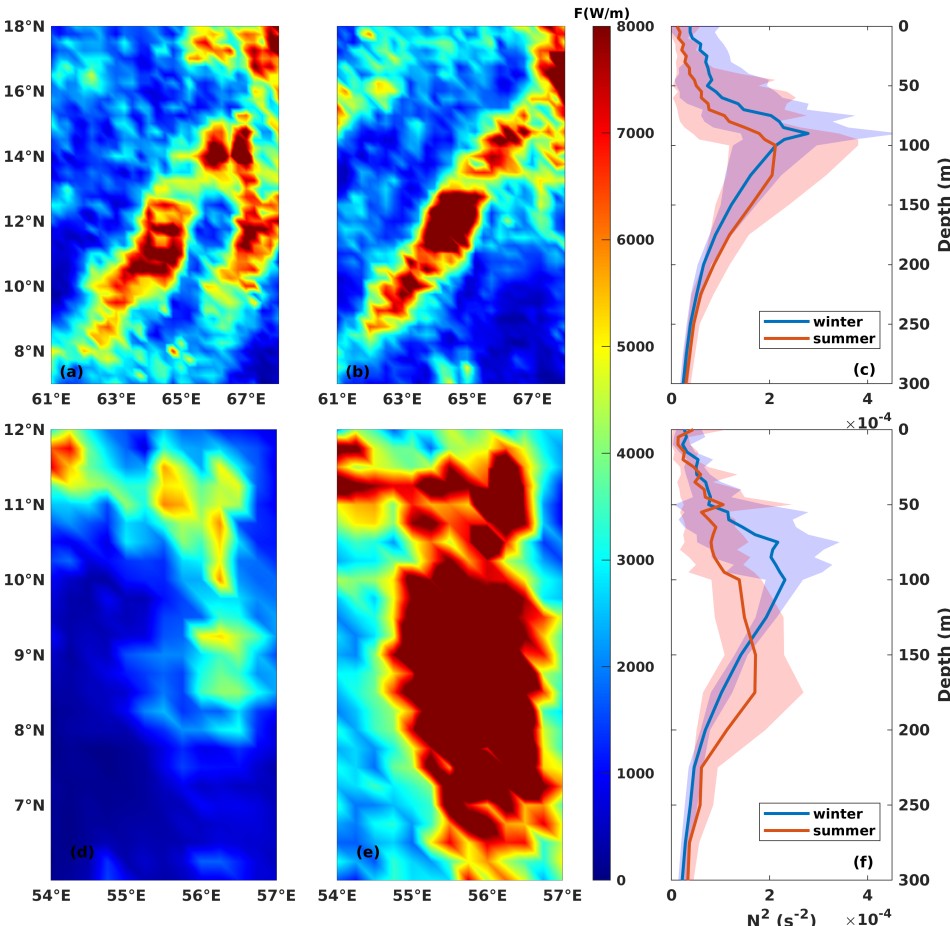

**Figure 8.** Zoomed-in vertically integrated energy flux in summer (**a**,**d**) and winter (**b**,**e**). (**c**,**f**) The domain-averaged seasonal $N^2$ profile in the upper 300 m. The blue and red colors represent the winter and summer, respectively. The shading area is the 95% confidence interval. (**a**–**c**) show the result of the black box in Figure 7, and (**d**–**f**) show the result of the orange box in Figure 7.

Similar to the SSH shown in the seasonal snapshots, the energy flux around the Carlsberg Ridge and Abd al Kuri Island is much higher in winter. To the north of Abd al Kuri Island, the maximum energy flux is estimated to be 15 kW/m both in summer and winter, while the large energy flux has wider coverage in winter. Similarly, internal tides around the Carlsberg Ridge exhibit significant energy flux. In the case of the Red Sea, the seasonal difference is not pronounced. In the Arabian (or Persian) Gulf, the primary seasonal difference appears at the Strait of Hormuz, where the energy flux is more significant in winter. Apart from these generation locations, internal tides close to the coastline also exhibit significant seasonal variations. For example, along the coastline of the Gulf of Oman, the energy flux in winter is stronger than that in summer.

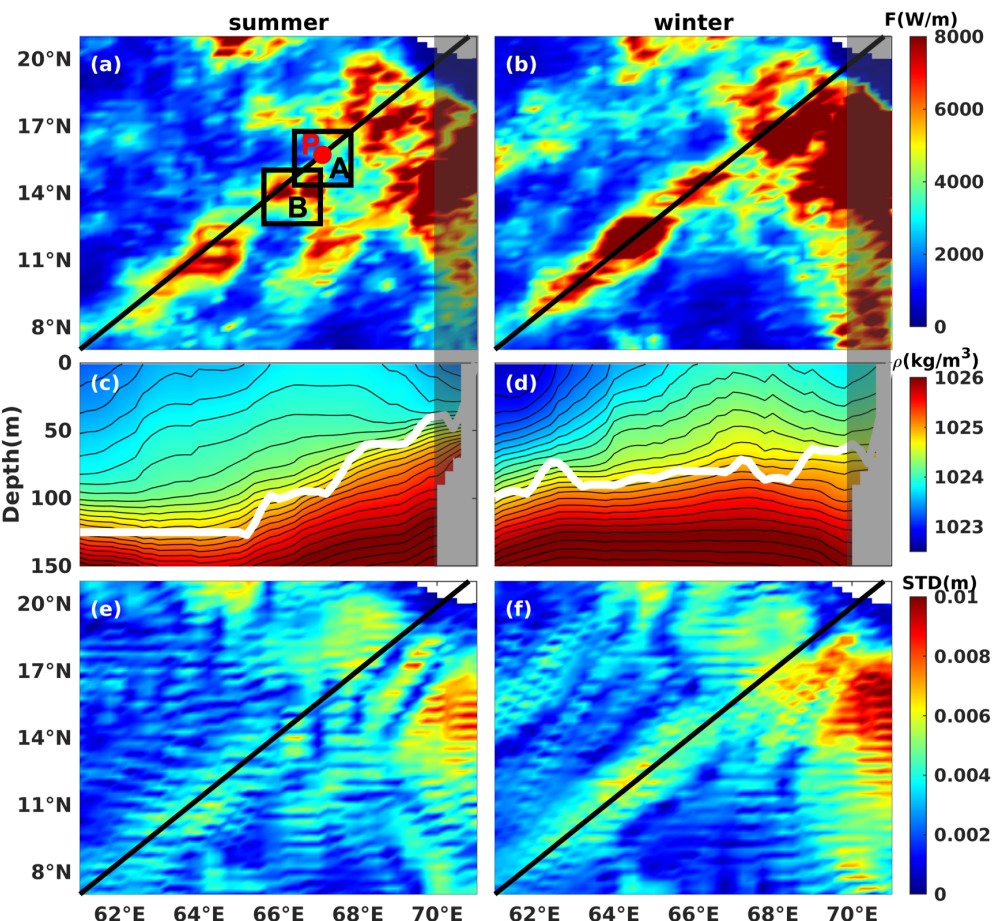

**Figure 9.** (**a**,**b**) Zoomed-in vertically integrated energy flux in summer and winter, respectively. The black line along the diagonal suggests the main propagation direction of internal tides. (**c**,**d**) display the corresponding density's cross-sections along the diagonals in (**a**,**b**). The white line suggests the depth of maximum $N^2$. The shaded area marks the range of the continental shelf. (**e**,**f**) are the standard deviation of the SSH.

## 4. Discussion

Our analysis suggests that the vertically integrated internal tide energy flux exhibits strong seasonal variability. Three factors may influence the internal tides: bathymetry, barotropic tides, and stratification [43,44]. Stratification is one of the most critical conditions controlling the generation of internal tides, which is usually assessed by the Brunt–Vaisala frequency $N$, and can modulate their seasonal variability [14,45]. Therefore, we examined the profiles of $N^2$ in different seasons in the area exhibiting significant seasonal variability in energy flux.

Figure 8a,b depict the summer and winter energy fluxes in the black box of Figure 7, and the corresponding domain-averaged $N^2$ is shown in Figure 8c. Figure 8d–f shows the behavior of the same variables around the Carlsberg Ridge; here, outstanding seasonal variability is observed. Surprisingly, compared to the difference in energy flux, the difference between the vertical distributions of $N^2$ in the two seasons is not very significant. In Figure 8f, the red shaded area is straighter than the blue one. This difference is caused by the seasonal variability of the Somalia current, which follows a northeast direction with a higher speed in summer and a southwest direction in winter [46]. In this situation, the southward internal tides generated around the Carlsberg Ridge meet the strong northward Somalia current during summer, thereby weakening the internal tides in this area. Simultaneously, the southward Somalia current strengthens the winter internal tides. The high-speed current may produce a large Doppler shift $\omega_0 - U_{mc}k$, where the subscript



*mc* refers to the mean current [47]. When the background current and waves have opposite directions, the signals can be suppressed [12]. In addition, the strong currents can increase the mixing of the ocean, which would weaken the stratification, as well as suppressing the strength of internal tides.

Doppler shifting also contributes to the generation of internal tides in the northeast Arabian Sea, where seasonal currents are present. However, according to the circulation described by Lachkar et al. [46], the current speed in the northeast Arabian Sea is much slower than that around Abd al Kuri Island, suggesting the existence of other influential factors. We then analyzed another aspect of internal tides with respect to horizontal variations in stratification. The energy flux from the area within the white boxes in Figure 7 is plotted in Figure 9a,b. Figure 9c,d illustrates the density distribution at the cross-sections, which are indicated by the black lines in Figure 9a. This black line was plotted to show the central axis along which energy propagates throughout the year. Theoretically, the internal tides generated under strong stratification are generally amplified [48]. Stratification in summer is stronger in the source region of internal tides (as indicated by the shading in Figure 9a–d); however, in this region, the density contours are closer in summer than in winter, and we observed only a weak energy flux in Box A (Figure 9a) in the propagation direction.

The weak energy flux in Box A may be attributed to the superposition of waves traveling from different directions, as reported by Zhao and Alford [23]. Our results indicate that the northeast Arabian Sea shelf is an active generation area of internal tides. The winter energy flux in Figure 9b also supports this finding. In addition, the red patch in Box B (Figure 9a) suggests that the energy must have originated from the shelves and subsequently propagated through Box A. Simultaneously, internal tides generated from different source locations meet in Box A, and after interacting, these waves propagate along different directions, moving away from each other. To visualize the interaction process, the changes in waves at an arbitrary point (Point P in Figure 9) are shown in Figure 10a. Waves 1–3 represent the three internal waves traveling in different directions at Point P. The solid lines depict the sum of the three waves. Because of the phase differences, these waves interact with each other, particularly in summer. The sum of the wave height in summer is markedly smaller than the individual heights of any of the three waves. In Figure 10b, Waves 1–3 are almost equal in amplitude, but different in phase, and the height of the winter sum wave is larger than those of Waves 1–3. The propagation and superposition of waves in the whole domain can be observed using more of the consecutive hourly SSH snapshots (not shown here). Moreover, the standard deviations (SDs) of those snapshots are shown in Figure 9e,f, indicating the deviation from the averaged SSH at each point. The SD distribution is similar to that of the surface amplitude of the internal tides in space, and a lower SD may reflect an offsetting of the waves.

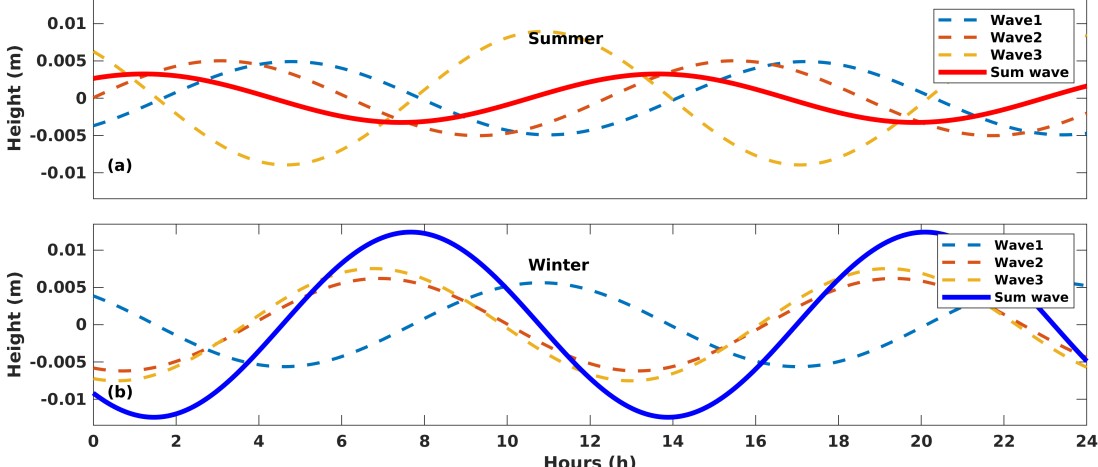

**Figure 10.** The 24 h time series of waves at Point P in summer (**a**) and winter (**b**). The dashed lines represent the three waves at Point P. The solid lines are the sum of the three.

After passing through Box B along the main axis, the energy flux in summer was lower than that in winter. The reduction in the energy flux in summer is not caused by the superposition of waves, because the internal waves in this region propagate in the same direction and are not likely to offset each other (as indicated by Figure 5b). A plausible reason for this seasonal difference in energy flux is the horizontal variations in stratification in the propagation direction of the internal tides. Several previous studies suggested that variations in stratification may affect the generation of internal tides. Lamb [49] investigated the sensitivity of internal tides to changes in stratification through numerical experiments, and Gerkema and van Haren [50] demonstrated that variations in stratification could change the internal tidal field. The white curves shown in Figure 9c,d depict the depth of the maximum $N^2$ in summer and winter, respectively, which varies by 90 m and 30 m within 1000 km, respectively. The pycnocline shows a clear slope in summer, whereas it is flat in winter. When the pycnocline has a slope (in summer), which means the the stratification is not stable in the spatial domain, the low-mode internal tides tend to cascade energy to the higher modes and are more likely to trigger the dissipation of turbulent kinetic energy, which hinders the propagation of internal tides [51]. In contrast, the flat pycnocline in winter provides more stable conditions for the propagation of internal tides over long distances. This inference is applied in Figure 9a–d: along the black line, areas with a low-energy flux correspond to the slopes at the white line. More work is still needed to understand the reasons behind this small energy flux.

## 5. Conclusions

In this study, we examined the general properties of internal tides and their seasonality in the Arabian Sea, including the Red Sea and the Arabian (or Persian) Gulf. Using datasets obtained form eight satellites, four major source locations of mode-1 $M_2$ internal tides in the study domain were identified: Abd al Kuri Island, the Carlsberg Ridge, the northeastern Arabian Sea, and the Maldive Islands. Internal tides near Abd al Kuri Island propagate meridionally, whereas those in the northeastern Arabian Sea mainly propagate to the southwest and the south. In terms of energy, the depth-integrated energy flux calculated from the annual data can exceed 10 kW/m in the northeastern Arabian Sea; this is a significant value, even from a global perspective. Moreover, in the Red Sea, we identified three source sites located in BAM, the southern Red Sea and the northern Red Sea. In the Arabian (or Persian) Gulf, internal tides were generated near the Strait of Hormuz and northern Qatar. In general, in decreasing order of intensity of internal tides, the source areas are the Arabian Sea, the Red Sea, and the Arabian (or Persian) Gulf.

We also analyzed the marked seasonal variability in the energy flux of the internal tides in the study domain, particularly around the Carlsberg Ridge and the northeastern Arabian Sea. Overall, the maximum energy flux of the internal tides is comparable in winter and summer, but the area with a high-energy flux is larger in winter. Around the Carlsberg Ridge, internal tides in winter are more pronounced in terms of both strength and propagation distance in all directions. Two branches of internal tides are observed to originate in the northeastern Arabian Sea shelf. Branch 1 of these internal tides propagates more regularly in winter, but these are mixed with waves from other directions in summer. The seasonal difference in the Arabian Sea is attributed to the Doppler shifting caused by background currents and horizontal variations in stratification in the propagation direction. The variation in stratification resulting from the different circulation patterns may favor the rising and suppression of isopycnals, which may further hinder the propagation of internal tides.

**Author Contributions:** Conceptualization, J.M. and D.G.; methodology, J.M. and D.G.; software, J.M.; validation, J.M.; formal analysis, J.M., D.G., P.Z., and I.H.; investigation, J.M.; resources, J.M.; data curation, J.M.; writing—original draft preparation, J.M.; writing—review and editing, J.M., D.G., P.Z., and I.H.; visualization, J.M.; supervision, I.H.; project administration, I.H.; funding acquisition, I.H. All authors read and agreed to the published version of the manuscript.

**Funding:** This research was funded by the Office of Sponsored Research (OSR) at King Abdullah University of Science and Technology (KAUST) under the Competitive Research Grants (CRG) program (Grant # URF/1/3408-01-01).

**Data Availability Statement:** The Cryosat-2 data were produced by Ssalto and distributed by AVISO (https://aviso.altimetry.fr, accessed on 19 March 2019). The WOA data were provided by NOAA (https://www.nodc.noaa.gov/OC5/woa18/, accessed on 19 May 2019). The data supporting the analysis presented here can be found at Figshare (https://figshare.com/articles/dataset/internal_tides_satellite_AS/14673195, accessed on 25 May 2021).

**Acknowledgments:** The authors would like to thank the King Abdullah University of Science and Technology for overall support to complete the research study. We thank all the researchers, institutions, and organizations responsible for making the WOA and AVISO datasets publicly available.

**Conflicts of Interest:** The authors declare no conflict of interest.

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
