# Peer review of "Seasonal M2 Internal Tides in the Arabian Sea"

_remotesensing, doi:10.3390/rs13142823_

Round 1

Reviewer 1 Report

  1. Is the analysis method same as that used by Dr. Zhao and colleagues, or the authors have developed a new different analysis method? If it is the former case, the authors better give credit to Zhao et al by early saying that they are using Zhao method. If it is the latter case, the authors better clearly state the major differences. For example, Equations (3) and (5) are from Zhao's papers, but the authors do not say it clearly. May we call this plagiarism? I do not know.
  2. The results are not new at all. Edward and Zhao have presented global maps of M2 internal tides, which cover the Arabian Sea. The authors better compare their new results with previous models. In this study, the authors do not discuss errors or compare with in situ measurements. One may wonder if their results are reliable or not? Why do they both waste time to repeat the same work?
  3. Figure 5b shows the energy flux vectors. But it seems messy in both amplitude and direction. This feature suggests that there are large errors in the results. The authors better do something on this.
  4. Figure 6 is interesting. However, they might have even larger errors. Are the differences real seasonal signals or errors? The authors better carefully discuss them.
  5. Figure 7a and 9a are weird in the black box. Why the energy flux is not smooth along the SW waves? The authors mention that there are three waves, but it is hard to tell in this figure. It seems the continental shelf is the only source. 
  6. Figure 9c, 9d, the authors suggest the effect of spatially varying stratification, but just by description. Are there any previous studies support this point? 
  7. Overall, the seasonal analysis is useful. The introduction of the analysis method is not clear (old method or new method?). The lack of error analysis is a major flaw. There is no calibration or validation of the analysis method. 

Reviewer 2 Report

Review report on “Seasonal M2 Internal Tides in the Arabian Sea” by Jingyi Ma et al.

The author uses data sets obtained from eight satellites to study the general characteristics and seasonality of the tides in the Arabian Sea. This research is helpful for regional ocean physical dynamic phenomena. However, I found that many details in the manuscript were omitted by the author, especially in the steps and chapters of the method, the author needs to describe in more detail. The author can find the insufficient parts of the current manuscript from the following suggestions.

  1. Figure 1a: What is C2? What do the blue dots and line segments represent? The red box in Table 1a should also be drawn on Table 1b.
  2. Lines 115: Is it a standard deviation based on long-term data at the same location (meaning that each location has a different standard deviation)?
  3. Line 117: Which item of equation (3) does the barotropic component refer to?
  4. Does the description in lines 116 to 119 refer to equation (3)? If so, please add the physical meaning or name of each parameter under equation (3).
  5. Line 126: Does wavelength (k) refer to the 50km and 120km mentioned in line 123?
  6. The phase speed (c) is estimated by formula (2). What are the values of these parameters? How is the measured profile data in formula (2) obtained? So, is c also a fixed value? What is the value of c when K=50km and k=200km?
  7. The meaning of lines 131 to 133 is not clear. What is the relationship between equation (1) and the blue dot and blue line in Figure 1a?
  8. Do lines 137 to 139 refer to equation (3)?
  9. Line 142: What are the considerations for using 1.6◦ × 1.6◦ window?
  10. Lines 147 & 156-157: How are amplitude and phase calculated?
  11. Line 148: How is "Time (t)" defined? What is the value?
  12. Lines 148-149: The range is 360 degrees. How many intervals does the angle calculation have?
  13. Lines 149-151: The first three largest internal tides (i = 3) were extracted, which means that formula (3) is the addition of three ai cos(). I strongly recommend that the author write clearly and carefully explain the value used for each (i=3).
  14. Lines 162-164: This sentence means:

wave 2 (Figure 2b) = SSH (Eq. (3) & Figure 2a)-Eq. (4);

wave 3 (Figure 2c) = wave 2-Eq. (4), right?

Here comes the question again. The above process uses equation (4) twice. What are the values in the equation during the two calculations? How is it calculated?

  1. Lines 166-167: The meaning of the whole sentence is not clear, what is 0.25 degree overlap? What is the weight?
  2. From lines 125 to 175, readers cannot use the data to calculate values according to the author’s simple instructions. In short, what are the steps of the author from obtaining the original data to drawing Figure 3? If possible, drawing a flowchart is also more helpful. What is the original data required?
  3. Lines 182-183: In Figure 3, the SSH in each location is the same time?
  4. Lines 188-189: How to judge the origin of internal tides from the Figure 3? What are the physical meanings of the positive and negative SSH values in Figure 3?
  5. Lines 215-216: Please explain in more detail why it was suppressed?
  6. Lines 217-218: How are these three positions defined?
  7. Lines 220-221: It is best to draw and add in the appendix.
  8. Lines 225-226: Does “five times” refer to the value of SSH?
  9. Line 227: How is 300 kilometers estimated?
  10. Line 236: Figure 3 only shows SSH, so how do the phase and amplitude of Figure 4a, b, d, and e be calculated? Is there a spatial distribution diagram like Figure 3 also available?
  11. Figure 5b is calculated from Equation 5? Why is it a vector with directionality? What is "a" in Equation 5?
  12. Figure 6 to 10: When are the snapshot images in summer and winter? Or is it a long-term average?
  13. Line 337: Add the formula of Brunt-Vaisala frequency N.
  14. Figure 10: Please extend the Y-axis in winter, the line segment is not completely displayed.

Round 2

Reviewer 1 Report

I think the revised manuscript is acceptable. 

Author Response

Nothing needs to reply.

Reviewer 2 Report

Dear authors,

The author gave a detailed response and solved almost all problems. There are only two small details left:

Reply to Response 6: How did the author obtain the vertical structures of displacement and vertical velocity data? Φ(z) will also be used in Equation 4. The data value which calculated by the author should be displayed in the manuscript.

Reply to Response 26: Figure 6 is a snapshot, so the detailed date and time should be given.
